# Pain-Related Brain Connectivity Changes in Migraine: A Narrative Review and Proof of Concept about Possible Novel Treatments Interference

**DOI:** 10.3390/brainsci11020234

**Published:** 2021-02-13

**Authors:** Marina de Tommaso, Eleonora Vecchio, Silvia Giovanna Quitadamo, Gianluca Coppola, Antonio Di Renzo, Vincenzo Parisi, Marcello Silvestro, Antonio Russo, Gioacchino Tedeschi

**Affiliations:** 1Applied Neurophysiology and Pain Unit, Bari Aldo Moro University, 70121 Bari, Italy; eleonora.vecchio@gmail.com (E.V.); silviagiovanna.quitadamo@uniba.it (S.G.Q.); 2Department of Medico-Surgical Sciences and Biotechnologies, Sapienza University of Rome Polo Pontino, Latina, 00185 Rome, Italy; gianluca.coppola@uniroma1.it; 3IRCCS—Fondazione Bietti, 00198 Rome, Italy; antoniomp777@hotmail.it (A.D.R.); vincenzo.parisi@fondazionebietti.it (V.P.); 4Clinica Neurologica e Neurofisiopatologia Università della Campania ‘Luigi Vanvitelli’, 81100 Napoli, Italy; marcello.silvestro6@gmail.com (M.S.); antonio.russo@unicampania.it (A.R.); gioacchino.tedeschi@unicampania.it (G.T.)

**Keywords:** migraine, pathophysiology, brain connectivity, FMRI, EEG, MEG, CGRP

## Abstract

A neuronal dysfunction based on the imbalance between excitatory and inhibitory cortical-subcortical neurotransmission seems at the basis of migraine. Intercritical neuronal abnormal excitability can culminate in the bioelectrical phenomenon of Cortical Spreading Depression (CSD) with secondary involvement of the vascular system and release of inflammatory mediators, modulating in turn neuronal activity. Neuronal dysfunction encompasses the altered connectivity between the brain areas implicated in the genesis, maintenance and chronic evolution of migraine. Advanced neuroimaging techniques allow to identify changes in functional connectivity (FC) between brain areas involved in pain processes. Through a narrative review, we re-searched case-control studies on FC in migraine, between 2015 and 2020, by inserting the words migraine, fMRI, EEG, MEG, connectivity, pain in Pubmed. Studies on FC have shown that cortical processes, in the neurolimbic pain network, are likely to be prevalent for triggering attacks, in response to predisposing factors, and that these lead to a demodulation of the subcortical areas, at the basis of migraine maintenance. The link between brain dysfunction and peripheral interactions through the inhibition of CGRP, the main mediator of sterile migraine inflammation needs to be further investigated. Preliminary evidence could suggest that peripheral nerves inference at somatic and trigeminal levels, appears to change brain FC.

## 1. Introduction

### Current Knowledge about Migraine Pathophysiology

The 2013 Global Burden of Disease Study reports migraine to be the sixth highest cause of years lost due to disability worldwide [1]. Chronic migraine—migraine more than 15 days of the month—affects 2% of the world population [2]. The economic costs of migraine, driven mainly by chronic migraine, range between $20 and $30 billion a year in the US [3].

Nowadays, the pathophysiological perspective outlines neuronal dysfunction as a predisposing factor to migraine attack onset, with secondary involvement of the vascular system. The cortical excitability of the migraine brain is different from the not migraine one, with a critical unbalance between excitatory and inhibitory neurotransmission, fluctuating across migraine phases. A genetic predisposition to such brain dysfunction [4,5], could facilitate the attack onset. It is unclear how a typical migraine attack is triggered—this is one of the most important unanswered questions in migraine neuroscience. Precipitating factors are environmental, as light, sounds and odors, emotional, or endogenous, depending on the general state of health. The triggers likely vary between and within subjects [6], depending on pre-existing network characteristics that might be quite individual. A thalamus-cortical dysrhythmia could be the pivotal abnormality causing the altered response to the multimodal stimuli [7]. The role of hypothalamus in determining abnormal adaptation to environmental conditions has been recently recognized [8]. Different migraine phases, the inter-critic, prodromal, acute and post-acute, are the expression of the fluctuation of brain functional state, which could be measured with different analysis [9]. Altered connectivity within cortical-subcortical circuits [10], defined an important dysfunctional aspect of migraine, with different connections within the default mode network [11], as well as the visual and pain-related cortical complex [12,13]. The complex interictal dysfunction of migraine brain, could culminate in the phenomenon of Cortical Spreading Depression (CSD). Since the pivotal study of Hadjikhani et al. [14], CSD is considered to be the physiological substrate of the migraine aura, and even of migraine without aura, that causes the spreading of a self-propagating wave of cellular depolarization in the cerebral cortex [15]. This bioelectrical phenomenon is followed in a causal mode by the activation of the trigemino-vascular system, with antidromic recruitment of meningeal vessels, vassal dilation and inflammatory neurotransmitters extravasation. This auto-induced sterile inflammation stimulates in turn trigeminal caudal nucleus and cervical C1–C3 roots [16]. The painful inputs from a-delta and C trigeminal and cervical afferents, reach the VPL and PO thalamus nuclei, the primary and secondary somatosensory cortex, the insula and the associative cortex, comprised in the so-called pain matrix or salience matrix network [17]. Calcitonin Gene-Related Peptide (CGRP), Substance P and nitric oxide are the main substances in the inflammatory process. The antagonistic action on CGRP receptors is thus a mode to cause an interruption of headache generation via a peripheral inhibition of the nociceptive system. Craniofacial afferents that synapse in the trigemino-cervical system project, directly or indirectly, to structures involved in the sensory/discriminatory, salience/alerting, and affective/motivational aspects of pain, as well as to structures involved in the response to pain-reflex autonomic and descending facilitatory/inhibitory modulation [18]. A dysfunction of descending pain modulation system, and probably basal hyperactivity of nociceptive neuron within the trigeminal caudal and C1–C3 nuclei, facilitate central sensitization phenomenon, allodynia generation, and neuronal plastic changes that render migraine patients prone to headache maintenance and persistence in a chronic mode [19]. The great challenge for the migraine approach is the understanding of the theoretic basis of the long-term effect of peripheral interaction via CGRP inhibition on the complex central dysfunction which could account for migraine inclusion within the “oscillopathies”. The complexity of brain function is based on dynamic relationships among cortical and subcortical areas, which enable the brain to adapt itself to different physiological and pathological conditions, a feature possibly affected in the migraine brain [20].

## 2. AIMS

This is a narrative review of recent studies on brain connectivity in migraine, focusing on fMRI and EEG/MEG studies reporting connectivity changes related to pain processing mechanisms.

This review intends to provide for a mechanistic hypothesis about the possible interference on the migraine dynamic model from the novel CGRP-related peripheral approach.

## 3. Methods

In this narrative review, we included studies based on PubMed search for migraine, fMRI, EEG, MEG, connectivity, pain in the 2015–2020 time window. We included only case-control studies, excluding case reports, reviews, animal models.

## 4. FMRI

### 4.1. Resting-State (Studies Concerning Pain-Related Brain Regions or Pain-Related Symptoms)

Since migraine is primarily a disorder of brain function and excitability, functional MRI (fMRI) studies represent a powerful tool to explore migraine neural correlates. In this frame, fMRI investigations fall into two main categories, specifically the event-related and the resting state ones, exploring respectively the brain activity in response to tasks and the connectivity of specific brain areas or networks at rest. The latter relays on several methodological approaches such as seed-based analysis, independent component analysis (ICA) and graph theory-based analysis [21].

Altogether, resting-state (RS) advanced neuroimaging techniques allow the identification of neuro-anatomical and biologically meaningful spatial brain maps organized in distributed and reproducible functional networks showing spontaneous and simultaneous fluctuations. 

### 4.2. Seed-Based Analysis Approach

The seed-based analysis is a model-based method looking for the linear correlation between a selected region of interest called “seed” and all the other voxels in the whole brain, thereby yielding seed-based FC maps. The seed-based analysis is a very good approach—due to its straightforwardness and interpretability—for exploring RS-fMRI functional connectivity (FC), especially when a strong “a priori” hypothesis is available. Accordingly, based on the pioneering advanced neuroimaging studies suggesting the involvement of specific brainstem areas in migraine attacks ignition, the so-called brainstem “migraine generator” has been assumed as a valuable “region of interest” in the seed-based investigations [22]. Therefore, disrupted pain-modulating descending pathways have been demonstrated in migraine patients and specifically abnormal FC has been identified between brainstem nuclei such as the periaqueductal grey matter (PAG) and the locus cuneiforms and between brainstem nuclei and several high-order cortical and subcortical regions (thalamus, posterior parietal cortex, anterior insula, frontal and temporal cortices) known to be involved in pain perception and modulation [23]. Interestingly, brainstem FC abnormalities in migraine patients seemed closely dependent on the presence of ictal cutaneous allodynia (CA) [24]. Altogether, the findings support dysfunctional top-down endogenous pain-inhibitory mechanisms, detectable also during interictal period, particularly relevant in migraine patients experiencing CA.

Nevertheless, the brainstem is only one of the subcortical structures that RS-fMRI investigations focused on to clarify the FC correlates underlying pathophysiological mechanisms. In this way, in the last decade, the hypothalamus has been widely investigated and its altered FC with specific brain regions involved in the regulation of sympathetic and parasympathetic functions (e.g., locus coeruleus, caudate, parahippocampal gyrus, cerebellum and temporal pole) has been demonstrated to be helpful in explaining some of the hypothalamic-mediated autonomic symptoms that accompany or precede migraine attacks [25]. However, in this scenario, the elegant studies of Schultze and May, which confirmed the key role of the hypothalamus in migraine pathophysiology and specifically showed increased hypothalamic activity in response to trigeminal nociceptive stimulation in the preictal phase of the migraine cycle (24–48 h before pain onset) along with an altered FC with both the spinal trigeminal nuclei and the dorsal rostral pons, should be considered a landmark in migraine FC studies. Interestingly, the RS-FC data collected from 9 migraine patients every day for 30 consecutive days also showed, still in the preictal phase, a higher FC between the nucleus accumbens and the amygdala, hippocampus and gyrus parahippocampalis compared to the interictal phase suggesting that dopaminergic changes may contribute to migraine attack generation and sustainment [26].

It is noteworthy that the thalamus constitutes the hub of central processing and integration of nociceptive inputs. In this context, several data have recently emerged on the role of altered thalamo-cortical FC patterns subtending multisensory integration abnormalities, deficits in cognitive attention, and evaluation of pain as well as in pain modulation in migraine patients. In particular, significantly weaker FC between the anterior dorsal thalamic nucleus and precuneus as well as between the ventral posterior nucleus and precuneus, inferior parietal lobule and middle frontal gyrus [27,28] has been observed in migraine patients.

To further complicate the complex migraine pathophysiology, very recently FC abnormalities have been identified between the cerebellum (e.g., crus I and crus II) and thalamus and other higher cortical areas [29] such as temporal, frontal, and parietal lobes as well as to secondary visual areas suggesting an abnormally reduced inhibitory involvement of the migraine cerebellum on nociceptive processing and evaluation [30].

In the context of subcortical structures, recent observations have supported the different role of the left or right amygdala in migraine pathophysiological mechanisms, suggesting the involvement of the left amygdala in migraine genesis as well as of the right amygdala in migraine chronicization. More clearly, a stronger FC between the amygdala and both the middle cingulate cortex and precuneus of the left hemisphere has been recently observed in patients with episodic migraine compared to healthy control (HC). Contrariwise, a stronger FC between the bilateral amygdala and widespread regions of the inferior temporal lobe, prefrontal gyrus, cingulate cortex and pre- and post-central gyri of the left and right hemispheres has been ascertained in patients with chronic migraine when compared to patients with episodic migraine. In another study, the amygdala, sensitized by means of repeated episodes of aura, may strengthen its connectivity with visceroceptive areas of the neurolimbic pain network (including insular and somatosensory cortices) [31]. The fact that the higher functional connectivity between the amygdala and viscerosensory areas characterizes migraine patients but not patients suffering from other chronic pain conditions not associated with cortical spreading depression (CSD) (e.g., trigeminal neuralgia and carpal tunnel syndrome) supports these findings as a possible “missing-link” between aura phenomenon and migraine headache. As can be easily argued, through the FC studies the old-fascinating but reductive idea of a single “migraine generator” was firstly replaced by the more extensive concept of a “migraine matrix” [32]. More recently, advances in RS studies showing FC abnormalities between brainstem nuclei of the so-called pain-modulating circuits and cortical and subcortical areas known to be involved in cognitive and affective attribution of pain experience and encompassing the limbic network, have suggested a more comprehensive model of migraine as the result of a dysfunctional “neurolimbic pain network” [33].

### 4.3. Independent Component Analysis Approach

The RS-fMRI ICA method relies on spontaneous low frequency fluctuations emerging from spatially separated, functionally linked, continuously communicating anatomical areas encompassing the so-called RS networks. Indeed, ICA facilitates the effective extraction from the background noise of distinct RS-fMRI networks by employing mathematical algorithms to decompose the signal from whole brain voxels to spatially and temporally independent components. The insights from the investigations of networks such as the sensory-motor network (SMN), default-mode network (DMN), central executive network (CEN), salience network (SN) and visual network (VN) in migraine patients, have shed light on the obscurity of the complex pathophysiology of migraine.

SMN-FC abnormalities including somatosensory (post-central gyrus) and motor (pre-central gyrus) regions (extending to the supplementary motor areas) have been consistently demonstrated in migraine without aura (MwoA) patients. Specifically, migraine patients exhibited significantly weaker FC between the primary somatosensory area and several cortical brain regions involved in pain intensity and spatial discrimination as well as in trigemino-thalamus-cortical nociceptive pathways [34,35]. It could be expected that SMN-FC abnormalities may represent the neuronal correlates of altered nociception pathways affecting the discrimination of pain sensory features in migraine patients.

By means of RS-fMRI ICA approach, a plethora of studies have been conducted to investigate the cognitive networks in order to understand the bidirectional relationship between pain experience and cognitive, affective and emotional dysregulation. The DMN is a well-known network highly relevant for self-referential cognitive processes, involved during stressful experiences, and coping strategies which are important to promote adaptation widely investigated in migraine patients, with apparently conflicting results so far [36,37]. Indeed, several investigations supported a decreased FC within the DMN anterior nodes (e.g., superior temporal gyrus and medial prefrontal cortices), all involved in perception and judgments of internal/external stimuli and in dealing with stressful experiences by the selections of appropriate adaptive and emotional responses in patients with MwoA. On the other hand, compelling evidence reported also an increased RS-FC in DMN posterior nodes (e.g., posterior cingulate cortex/precuneus) engaged in both pain sensitivity and integration of inputs from different sensory modalities (multisensory integration). Altogether, these findings support a functional differentiation between brain areas constituting the anterior and posterior DMN nodes engaged in an anti-correlated activity (e.g., lower the mPFC response, higher the PCC/precuneus response and vice versa) further suggesting a compensatory adaptive mechanism characterized by posterior DMN nodes support (or partial replacement) to the dysfunctional anterior DMN regions [38]. The failure of the above-mentioned adaptive postero-anterior compensatory mechanism, witnessed by a significantly weaker FC of the PCC/precuneus, which was shown in MwoA patients who will develop cutaneous allodynia (CA) over time compared to those patients who will not develop CA may justify an aberrant multisensory integration leading to CA, increased vulnerability to stressors and, overall, a tendency to migraine chronic evolution. The concept mentioned above is further supported by previous observations showing the same abnormal RS-FC pattern in patients with MwoA who will develop CA and in chronic migraine patients as well as in patients suffering from other primary chronic painful conditions such as low-back pain and fibromyalgia.

The CEN is a cognitive network responsible for high-level cognitive functions and known to underlie neuronal correlates of executive function, a set of high-order cognitive processes involved in multiple aspects of individual daily living experiences such as planning for the future and adaptation (i.e., allostasis). In line with neuropsychological studies, showing a slight impairment of complex executive functions including the goals maintenance, the correct allocation of attentional resources, and performance monitoring in migraine patients, a reduced FC within the CEN (centered in the middle frontal gyrus and the anterior cingulate cortex) has been previously reported in both MwoA and migraine with aura (MwA) when compared to HC. Interestingly, precisely CEN-FC abnormalities have been recently demonstrated to predict the development of CA, working as putative advanced neuroimaging prognostic biomarkers able to identify those patients who will develop CA over time and, therefore, those patients more prone to chronic migraine as well [38].

The brain network that constitutes the so-called SN encompasses areas involved in the evaluation of specific inputs from external or internal behavior by assigning the appropriate relevance to stimuli for continuous processing and pain modulation. A decreased SN-FC centered in the anterior cingulate cortex, anterior insula, and parahippocampal gyrus has been consistently observed in migraine patients when compared to HC suggesting a dysfunctional mechanism leading to the excessive attribution of “salience” to normally non-salient stimuli [39].

Altogether, the observed abnormalities of RS-FC networks subtending adaptive behavior could lead to interpret migraine through the lens of maladaptive stress responses or as a model of “allostatic overload” brain disease. In other terms, stressors trigger migraine attacks but repetitive and frequent attacks can work as stressors themselves, generating a vicious cycle able to disrupt neuro-limbic pain networks and to induce migraine chronicity [40].

The high prevalence of photophobia during the attacks as well as the light hypersensitivity during the interictal period and the occurrence of visual stimuli as trigger factors in migraine patients have induced a growing interest in the visual pathway and, more recently on VN, in order to clarify whether FC abnormalities may be part of a constitutional pattern (likely aimed to an adaptive/defensive mechanism) or represent an acquired sensitization phenomenon in migraine patients. In this context, a significantly increased FC in the lingual gyrus within the VN in patients with MwA when compared to both patients with MwoA and HC has been found during the interictal period, supporting the hypothesis of an extrastriate cortex involvement in the genesis and propagation of aura phenomenon and the putatively subtending cortical spreading depression. On the other hand, even more fMRI studies support the visual cortex involvement also in migraine patients not experiencing the aura phenomenon [41]. More specifically, a disrupted FC has been found between the SMN and the visual cortices in a cohort of MwoA patients during migraine attacks induced by the administration of PACAP38 [42]. Similarly, a decreased FC between the DMN and the visuo-spatial system in episodic MwoA patients during the interictal period [43] has been observed. Nevertheless, a recent connectivity analysis in MwoA patients showed increased functional anti-correlation between the temporo-parietal cortex and the bilateral visual cortex [44].

RS-FC studies have been recently conducted to identify the link between the aura phenomenon and the trigemino-vascular complex activation (known to subtend the headache phase), an unsolved conundrum about MwA pathophysiology [45]. Among these, no differences have been observed in FC between visual cortex and brain areas known to be involved in pain perception and modulation, suggesting that neurophysiological correlates of aura phenomenon may be unable to activate trigemino-vascular system and, therefore, migraine pain is not a mere CSD consequence. However, given that patients were scanned during the headache phase when the aura was already concluded, we cannot exclude that a hypothetical CSD-induced signal to pain-processing structures, only occurring during the aura phenomenon, could have been missed.

In addition to the abnormal intrinsic connectivity of the main RSN, a disrupted large-scale cognitive network crosstalk has been demonstrated in chronic migraine patients. More specifically, chronic migraine patients showed a significantly reduced FC between the CEN and both the DMN and the dorsal attention network (DAS) and a significantly stronger FC between the DAS and the DMN [46]. Similarly, reduced FC has been demonstrated between several RS-networks and subcortical structures such as the thalamus and hypothalamus [47] supporting the involvement of these subcortical structures in large-scale network reorganization in migraine patients.

In a recent study, Schwedt and colleagues, looking for a putative FC migraine signature, demonstrated that migraine is associated with several highly reproducible FC alterations in different brain regions enough for constituting a brain-connectivity neuroimaging biomarker, to differentiate migraine patients from HC [48]. More specifically, FC parameters from six brain areas derived from previous RS studies (e.g., right middle temporal, posterior insula, middle cingulate, left ventromedial prefrontal and bilateral amygdala) were used to create a classification algorithm to determine whether a “specific FC brain map” is able to identify a migraine patient or a HC. Interestingly, this algorithm is characterized by an accuracy of detection between patient and HC of 86.1% strictly dependent on the brain circuitry reorganization due to disease duration (96.7% accuracy in patients with disease duration longer than 14 years).

### 4.4. Graph Theory Analysis Approach

In the last decades, driven by the ascendancy of network science, a novel RS-fMRI methodological approach emerged to explore the structural neural substrates enabling functional communication within the brain “connectome”. According to the graph theory analysis, the brain network is investigated as a set of grey matter regions (nodes) structurally connected by white matter paths (edges). Two distinct dimensions (along which brain connectome network is organized), known by the names of ‘segregation’ and ‘integration’, provide a general framework that allows the description and the categorization of different disorders. Specially, in line with previous electroencephalographic and magnetoencephalographic observations in migraine patients, increased global efficiency and clustering of cortical areas [12,49,50,51,52] have consistently been reported in episodic migraine patients. The first index reflects the network ability to manage a large flow of information while the second one describes the tendency of nearest nodes to connect each other constituting specialized modules. However, the increase of integration and clustering in both cortical networks may provoke an imbalance between the need to invest resources to promote network efficiency and segregation, and the need to minimize the physical and metabolic cost of wiring, due to critical abnormalities in energy metabolism (likely related to mitochondrial energy dysfunction). In this setting, migraine (subtended by a high energy consuming brain) could be one of the evolutionary prices that the human species had to pay to gain a developed and highly connected neocortex and, in turn, a highly performing central nervous system. In Table 1, studies published in the 2015–2020 period are summarized. (Table 1).

### 4.5. Pain Stimulation-Related FMRI Connectivity Changes

One of the most widely used methods to study brain activity in migraineurs is to send a controlled painful stimulus to the trigeminal region or to an extra-cephalic region, such as the back of the hand, and to analyze the evoked activity of the BOLD signal to the functional MRI. Both caloric stimuli and gaseous ammonia have been used in this regard. Compared to healthy subjects, the vast majority of studies carried out in migraineurs between attacks revealed an increase in BOLD activity of cortico-subcortical brain areas involved in noxious information processing and anti-nociception [53,54,55,56,57,58]. Most of these areas are anchored to the so-called pain neuromatrix, which nowadays is known as salience network, devoted to select the most relevant internal and external events [59], as for instance the unpleasant feeling of pain and the threatening sensation of receiving pain.

In one study, the strength of the pain-induced BOLD signal changes did not correlate with the presence of psychopathology [60]. Mehnert et al. [61] specifically studies the cerebellar response to pain delivered over the nose by gaseous ammonia and found linearly increased activation of cerebellar areas with the sense of unpleasantness and in parallel with the activation of periaqueductal grey (PAG). Moreover, they observed an increased interaction during the delivery of pain stimuli between the cerebellum and several brain areas including the thalamus, PAG, basal ganglia, bilateral insula and lingual gyrus, supramarginal and precentral areas, all areas devoted to pain perception and anti-nociception. The subjective perception of headache intensity was negatively related to the middle prefrontal cortex and posterior cingulate cortex (PCC) and positively related to the cerebellar [61] and bilateral insula activation [53].

Using gaseous ammonia as painful stimulation, researchers found that the interictal reduced BOLD activation strength within the brainstem (trigeminal nucleus caudalis) [8,58] was positively correlated with the time to the next attack [58]. In this respect, it is interesting to note that in migraine patients the trigeminal spinal nucleus within the brainstem is active even when the stimulus is not harmful, such as looking at a checkerboard [62]. This may mean that the activation of this system is the result of a non-specific response to any external stimulus that is perceived as unpleasant, as the aversion to visual stimuli in the absence of eye, retinal or optic nerve diseases. Attack frequency was related to activation strength within several brain areas belonging to the pain neuromatrix, such as middle DLPFC, precentral gyrus, cingulate cortex, fusiform gyrus, insula, hippocampus, PAG, and cerebellum [29,53,54,60]. The duration of the history of migraine was correlated positively with Blood Oxygen Level Dependent (BOLD)activation strength in the fusiform gyrus [60] and with that in the cerebellum [55], and negatively with that in the superior temporal gyrus [53].

The functional BOLD activity related to the pain stimulus was also used to investigate the pathophysiology of the migraine attack and the hours immediately preceding it.

In a series of patients affected by episodic migraine scanned every day for 1-month, the hypothalamus was significantly more activated during the 48 h preceding the beginning of a migraine attack [8,63]. During the attack, the hypothalamus was functionally coupled with the spinal trigeminal nuclei and with the dorsal rostral pons [63], an area classically considered to be a possible migraine “generator” [58,64,65]. It comes as an odd that, as can be seen from the images accompanying the article and as, in part, described by the authors themselves, the primary visual areas appear to be active during both the pre-ictal and ictal phases [8,63], as already suggested by others using PET scanning instead [66]. All this spatial evidence seems to be a complement to the evidence obtained previously with electro-functional methods, i.e., an increased visual cortex activity during the pre-ictal phase and ictal phase coupled with a brainstem activation [67,68,69].

During the attacks, sometimes compared with the interictal phase other times with the responses from healthy controls, a greater BOLD signal was detected in response to noxious stimuli within the temporal lobes [54], the dorsal pons [58], the posterior hypothalamus [70,71], and the cerebellum [61]. Notably, the spinal trigeminal nucleus activation seems to fluctuate during the migraine cycle; it was slightly greater during the preictal state in migraine patients than in healthy subjects, but it was significantly increased in the preictal state compared to the response obtained between attacks. During an attack, the activation of the spinal trigeminal nucleus diminished as compared to the preictal period [58], at a time when it was more clearly activated by visual sensory load [62]. This evidence points once again to an important role played by the visual system in the pathophysiology of migraine attack recurrence.

In the Table 2, studies specifically reporting functional connectivity changes induced by noxious stimulation are summarized (Table 2)

### 4.6. Effects of Preventive Drugs

There are only a few papers exploring the effects of drugs used for migraine therapy on cortical processing of pain information (see Table 2).

Some researchers observed that 2 to 3-month treatment with the beta-blocker metoprolol did not induce any major different BOLD activation on the trigeminal pain processing of patients with migraine. However, using a more liberal and exploratory statistical threshold, the BOLD signal intensity within the hypothalamus increased during the delivery of pain stimuli compared to that under placebo suggesting a more peripheral effects of metoprolol with negligible effects [72] 

A recent fMRI study of the same group of researchers challenged this interpretation. Indeed, they explored the effects of 2-weeks treatment with Erenumab, a monoclonal antibody against the CGRP receptor, on central pain processing and found decreased BOLD activation within the thalamus contralateral to the stimulated side, of right middle temporal gyrus, right lingual gyrus, left operculum, and of both sides of the cerebellum [73] Therefore, in a secondary analysis, only those patients responding to Erenumab showed a significant reduction of the BOLD signal within the hypothalamus compared to the baseline.

## 5. EEG/MEG

The different way in which migraineurs process multimodal stimuli could be explored by methods with optimal time resolution [71]. The high temporal resolution of EEG/MEG enables the study of functional connectivity, which could support the effective and structural connectivity data based on magnetic resonance imaging (MRI). Methods such as correlations, spectral coherence, and phase synchronization, demonstrate the extent of the statistical connection of two variables and reveal functional connectivity. The properties of EEG/MEG signals also enable the evaluation of the flow of connections and information across different brain areas. This permits the extension of functional connectivity by explaining the architecture of connections between two correlated time series. Methods, such as Granger causality, or biologically inspired, such as dynamic causal modeling, could shed light on the information flow in the brain intended as a nonlinear system [72,73,74]. The application of such methods to resting state EEG/MEG and EEG/MEG rhythm perturbation induced by multimodal and specifically somatosensory stimuli appears to be particularly adapt in describing the complexity of the migraine brain.

### 5.1. Resting State (Studies Concerning Pain-Related Brain Regions or Pain Related Symptoms)

A study applied effective connectivity using nonlinear Granger Causality (GC) and brain networking analysis to 65 channels basal and visual stimulation related EEG in 19 migraine with aura (MA), 19 without aura (MO), and 11 controls and compared these findings with signal changes. A different pattern of reduced vs. increased GC respectively in MO and MA patients, as compared to controls, emerged in resting state specially in the occipital cortex, outlining a different pattern of connections and information exchange in the parietal-occipital regions, confirmed by the enhanced occipital bold signal in MA patients during visual stimulation [12].

A more recent study confirmed that the EEG-based resting-state connectivity is different between migraine with and without aura, as patients with aura had overall lower connectivity within the theta band in the cortical areas within the Default Mode Network (DMN) and Resting State Networks (RSN) [75].

Interestingly, EEG phase coherence fluctuated across migraine phases, as demonstrated by Cao et al. [76] in 50 migraine patients compared to 20 controls. In fact, compared to controls, EEG power and coherence were lower in inter-ictal and ictal patients, which were “normalized” in the pre-ictal or post-ictal groups.

One of the first MEG connectivity study in migraine described significantly increased functional connectivity in the slow wave (0.1–1 Hz) band in the frontal area as compared with controls, with migraine with aura showing increased functional connectivity in the theta (4–8 Hz) band in the occipital area. The networking structure appeared also different in migraineurs as compared to controls, indicating an abnormality of functional connectivity in both low- and high-frequency ranges [77].

Recently, Nieboer et al. [78], estimated functional connectivity between 78 cortical brain regions by calculating the phase lag index, in 24 healthy controls and 24 migraine patients, divided into 2 groups for short or long disease duration. Altered networking patterns in higher frequency bands characterized patients with longer disease duration, confirming long term effect of migraine disease on cortical functional architecture rearrangement.

A MEG study evaluated possible correlations between individual pain sensitivity and functional connectivity at 2 to 59 Hz in pain-related cortical regions, bilateral anterior and posterior cingulate cortex, medial and lateral orbitofrontal (MOF) cortex, insula, the primary somatosensory cortex (SI), the primary motor cortex (MI) in 30 migraine patients and 27 controls. Migraine patients showed a lack of correlation between pain sensitivity and resting-state gamma oscillation, present in the control group, for a possible disruption of pain-related cortical networks [79].

### 5.2. Pain Stimulation-Related EEG/MEG Connectivity Changes

A new neurophysiological interpretation of habituation phenomena in terms of progressive synchronization and reduction of information flow within the neuronal networks activated by repetitive stimuli was suggested [13]. Following painful laser stimulation delivered to the right hand, EEG rhythms exhibited lively information flow, as measured by Granger causality, in migraine patients compared with controls, who went into a progressive synchronization. The rate of information flow was inversely correlated with the habituation of averaged laser evoked responses. This correlation suggested that the phenomenon of progressive adaptation to external conditions could reduce the need for cortical connections between distant regions, and create synchronized networks with reduction of stress and energy demand.

A more recent study investigated modulations in effective connectivity between the sources of laser evoked potentials (LEPs) from a first to the final block of trigeminal LEPs using dynamic causal modeling (DCM) in a group of 23 migraine patients and 20 controls. The connections between the secondary somatosensory areas (lS2, rS2), insulae, anterior cingulate cortex (ACC), contralateral primary somatosensory cortex (lS1), and thalamus were investigated. Migraine patients show amplified relationships between lS1 and the thalamus, in both directions, with a lack of progressive habituation of connection strengths from the thalamus to all somatosensory areas, in line with altered thalamo-cortical network dynamics [80].

Ren et al. [81] used MEG to detect the possible change in functional connectivity during median nerve stimulation in the interictal state of 22 migraine patients and 22 controls. Migraine patients exhibited increased functional connectivity in the high-frequency (250–1000 Hz) band between the sensory cortex and the frontal lobe, and several brain networking abnormalities in all EEG rhythms, expressed by the graph theory application. Authors concluded with the hypothesis of aberrant connections from the somatosensory cortex to the frontal lobe, with increased heighten of cortical networks. (Table 3).

Summarizing, EEG and MEG connectivity results converge toward disrupted network connections in resting state and under painful stimulation in migraine patients, while some differences emerged within migraine phenotypes (with and without aura) and migraine phases.

## 6. General Remarks

In conclusion, although advanced neuroimaging investigations surely represent a valuable tool to improve our knowledge of migraine pathophysiological mechanisms, a pathophysiological interpretation strongly depends on whether we would disentangle mechanisms underlying migraine attack or migraine disorder. Indeed, the lesson we have learned from Functional Connectivity studies is that subcortical structures such as the hypothalamus, thalamus or brainstem are surely involved in some moments of the migraine cycle such as the prodromal or headache phases. Moreover, migraine appears a pervasive condition, likely with a prevalence of cortical processes, leading to migraine attacks susceptibility in response to several and different predisposing factors.

Indeed, changes in the activity of different cortical brain regions encompassing the neurolimbic-pain network, secondarily allowing a de-modulation of subcortical areas such as hypothalamus, amygdala and brainstem nuclei in a continuous mutual cross-talk, may represent the “neuronal background” of the migraine brain. In other terms, abnormal cortical responsivity and sensory processing may constitute the fingerprint of the migraine brain and, in the course of sensory overload and lowered energy reserve, it may ignite the major pain signaling system of the brain, the trigeminovascular system, inducing migraine attacks. In the latter, subcortical brain trigeminal and extra-trigeminal structures seem to manage the attacks propagation and maintenance.

## 7. Perspectives (in View of Peripheral CGRP Approach)

A new era for migraine treatment was opened with the use of large molecules acting on CGRP, the main mediator of sterile inflammation. The CGRP-mAbs act at the peripheral level, with an improbable and weak effect on the CNS [82], which would reduce their long term efficacy on the complex migraine brain dysfunction. However, a very recent study reported that a single dose 70 mg of enerumab caused a relevant change of functional connectivity from the hypothalamus to the insula, temporal lobe, hippocampus and trigeminal nuclei. In addition, responders showed a specific reduction of hypothalamus activation [83]. Authors thus supposed a direct central effect of erenumab, largely questioned by basic experiments [53], but reliable based on the preventive effect on migraine attack number. However, the change of peripheral input modality, due to primary peripheral nervous system pathology or treatment interference, have a strong central effect with plastic re-adaptation and changes in functional connectivity modalities.

In the present scientific scenario, growing evidence are supporting how peripheral nerves involvement at somatic and trigeminal level, could change brain functional connectivity, specially within the network devoted to pain processing [84,85]. The peripheral action of sumatriptan, changed network-level functional connectivity in the inflammatory soup model of migraine-like pathophysiology induced in conscious rats [86].

Animal experiments indicate that CGRP or activation of its receptor(s) may contribute to CSD propagation. There would be a bi-directional relationship between vascular and neuronal activities, where CSD activated changes in C-fibers function with CGRP antidromic production, while the subsequent change in vascular tone, at least in vitro, can, in turn, modulate neuronal activity, termed vascular-neuro coupling [87]. Mathematical models of CSD on realistic MRI from migraine with aura patients showed that the propagation of hyperpolarization phenomenon could in theory involve cortical areas in the network of pain processing [88]. The tonic inhibition of trigeminal peripheral afferents inputs to the cortex, exerted by CGRP monoclonal antibodies, could thus modulate the excitability of the large network dedicated to nociceptive signals elaboration, also changing the mode of cortical connections favoring the generation and extension of critical bioelectrical phenomena.

The complex neuronal dynamic architecture subtending migraine is worthy of an integrated approach toward this complex disease. However, future long term efficacy studies on CGRP-mAbs, coupled with neurophysiological evidence of possible interference on brain connectivity features, could possibly confirm the impression that a specific and prolonged action on the trigemino-vascular system, though limited to the periphery, has a pivotal role in migraine puzzle resolving.

## Figures and Tables

**Table 1 brainsci-11-00234-t001:** Case Control Resting State Functional Connectivity Investigations.

Year	Authors	Subjects	Methods	Condition	Main Results (in Pts)	Significance
2015	Liu et al. [31]	108 MwoA 30 HC	Graph theory analysis approach	Intercritical	Significant between-group differences in the intensity of the brain connections	Learning mechanisms are likely involved in maintenance of chronic migraine pain
2015	Tessitore et al. [22]	20 MwA 20 MwoA 20 HC	RS-FC BOLD-fMRI; ICA- approach	Intercritical	Reduced RS-FC within the CEN centred in middle frontal gyrus and anterior cingulate cortex in both MwoA and MwA pts	Vulnerability to executive high-demanding conditions of daily living activities in pts with MwoA and MwA
2015	Hougaard et al. [43]	40 MwA 40 HC	RS-FC BOLD-fMRI; Seed-based analysis approach	Intercritical	No intrinsic FC abnormalities in the intercritical phase of migraine with aura	MwA brain may be abnormally functioning in intercritical period only during exposure to external stimuli
2016	Tedeschi et al. [32]	20 MwA 20 MwoA 20 HC	RS-FC BOLD-fMRI; ICA- approach	Intercritical	Increased RS-FC of lingual gyrus within VN in MwA pts	Extrastriate cortex, centred in the lingual gyrus, play a critical role in mechanisms underlying the initiation and propagation of the migraine aura phenomenon
2016	Niddam et al. [34]	26 MwA 26 MwoA 26 HC	RS-FC BOLD-fMRI; ICA- approach	Intercritical	Reduced RS-FC between anterior insula and occipital areas in MwA pts	An unique pattern of connectivity involving area V3A, (known to be implicated in aura generation) characterizes MwA pts
2016	Zhang et al. [38]	22 MwA 22 HC	RS-FC BOLD-fMRI; ICA- approach	Intercritical	Increased RS-FC in precuneus/posterior cingulate cortex within the DMN and increased in ReHo values in bilateral precuneus/posterior cingulate cortex, pons and trigeminal nerve entry zone	Abnormalities in the precuneus/posterior cingulate cortex suggest a dysfunctional multimodal integration in MwoA pts
2016	Amin et al. [43]	24 MwoA	RS-FC BOLD-fMRI; ICA- approach	Ictal (PACAP induced attacks)	Increased RS-FC of SN, SMN and decreased RS-FC of VN during PACAP38 induced migraine attacks	PACAP38-induced migraine attack is associated with altered connectivity of several large-scale functional networks of the brain
2016	Coppola et al. [11]	18 MwoA 19 HC	RS-FC BOLD-fMRI; ICA- approach	Intercritical	Reduced RS-FC between DMN and visuo-spatial system	Abnormal connectivity within networks could contribute to migraine pathophysiology
2017	Chen et al. [31]	18 episodic MwoA 16 chronic migraine 18 HC	RS-FC BOLD-fMRI; Seed-based analysis approach	Intercritical	Increased RS-FC of left amygdala in episodic migraine pts and decreased RS-FC of right amygdala in chronic migraine pts compared with HC. Increased FC of bilateral amygdala in chronic migraine pts compared with episodic migraine pts	Altered connectivity of amygdala support neurolimbic pain network contributes in the episodic migraine pathogenesis and migraine chronification
2017	Zhang et al. [27]	30 MwoA 31 HC	RS-FC BOLD-fMRI; ICA- approach	Intercritical	Reduced RS-FC between S1 and brain areas within the pain intensity and spatial discrimination pathways and trigemino-thalamo-cortical nociceptive pathway	Decreased connectivity between the S1 and brain areas in migraine pts may disrupt the discrimination of sensory features of pain and affect nociception pathways
2017	Androulakis et al. [40]	29 women with chronic migraine 29 HC	RS-FC BOLD-fMRI; ICA- approach	Intercritical	Decreased RS-FC of DMN, SN and CEN	Decreased overall connectivity of the 3 major intrinsic brain networks in women with chronic migraine, correlated with frequency of moderate to severe headache and CA
2017	Li et al. [35]	30 MwoA 30 HC	Graph theory analysis approach	Intercritical	Altered rich club organization with high level of feeder connection density, abnormal small-world organization with increased global efficiency and decreased strength of SC-FC coupling	Selective alteration of the connectivity of the rich club regions which probably increases the integration among pain-related brain circuits with increased excitability but reduced inhibition in migraine modulation
2019	Tu et al. [28]	89 MwoA 70 HC	_d_RS-FC BOLD-fMRI; Seed-based analysis approach	Intercritical	Abnormal dynamic RS-FC of posterior thalamus (pulvinar nucleus) with the visual cortex and the precuneus, correlated with migraine frequency	MwoA pts are characterized by transient pathologic state with atypical thalamo-cortical connectivity
2019	Huang et al. [10]	30 MwoA 22 HC	RS-FC BOLD-fMRI; Seed-based analysis approach	Intercritical	Reduced RS-FC between both red nucleus and substantia nigra and several cortical and subcortical brain regions	The functional changes of regions associated with cognitive evaluation, multisensory integration, and pain modulation and perception may be associated with migraine production, feedback and development
2019	Coppola et al. [47]	20 chronic migraine 20 HC	RS-FC BOLD-fMRI; ICA- approach	Intercritical	Reduced RS-FC between DMN and CEN and between DAS and CEN, increased RS-FC between DAS and DMN	Large-scale reorganization of functional cortical networks in chronic migraine
2020	Schulte et al. [13]	12 MwoA (during intercritical, preictal and ictal phase)	RS-FC BOLD-fMRI; Seed-based analysis approach	Intercritical and ictal	High RS-FC between accumbens and amygdala, hippocampus, gyrus parahippocampalis and dorsal rostral pons in the preictal phase compared to the intercritical phase	Changes of connectivity in dopaminergic centres and between the dorsal pons and the hypothalamus are important in migraine attack generation and sustainment.
2020	Coppola et al. [48]	20 chronic migraine 20 HC	RS-FC BOLD-fMRI; ICA- approach	Intercritical	Increased RS-FC between the hypothalamus and brain areas belonging to the DMN and dorsal VN. No RS-FC abnormalities between hypothalamus and brainstem	Hypothalamic involvement in large-scale reorganisation at the functional-network level in chronic migraine
2020	Qin et al. [27]	48 MwoA 48 HC	RS-FC BOLD-fMRI; Seed-based analysis approach	Intercritical	Reduced RS-FC between anterior dorsal thalamic nucleus and precuneus and between the ventral posterior nucleus and precuneus, inferior parietal lobule and middle frontal gyrus	Altered thalamo-cortical connectivity patterns may contribute to multisensory integration abnormalities, deficits in pain attention, cognitive evaluation and pain modulation
2020	Wei et al. [36]	40 MwoA 34 HC	RS-FC BOLD-fMRI; ICA- approach	Intercritical	Decreased RS-FC between SMN and middle temporal gyrus, putamen, insula and precuneus, increased RS-FC between SMN and paracentral lobule	SMN intra- and internetwork connectivity imbalances associated with nociceptive regulation and migraine chronification
2020	Ke et al. [30]	39 MwA 35 HC	RS-FC BOLD-fMRI; Seed-based analysis approach	Intercritical	Increased RS-FC of posterior insula with the postcentral gyrus, supplementary motor area/paracentral lobule, fusiform gyrus and temporal pole, decreased RS-FC between crus I and medial prefrontal cortex, angular gyrus, medial and lateral temporal cortices	Increased connectivity with the posterior insula and decreased connectivity of crus I may underlie disturbed sensory integration and cognitive pain processing
2020	Russo et al. [22]	37 MwoA (with or without CA development) 19 HC	RS-FC BOLD-fMRI; ICA- approach	Intercritical	Reduced RS-FC of both DMN and CEN in pts with MwoA developing CA when compared with both pts with MwoA not developing CA and HC	Abnormalities in DMN and CEN connectivity could represent a prognostic imaging biomarker able to identify migraine pts more prone to experiencing CA and, therefore, more inclined to chronic migraine

Legend. BOLD: blood oxygenation level dependent; CA: cutaneous allodynia; CEN: central executive network; CSD: cortical spreading depression; CT: cortical thickness; DAS: dorsal attentional system; DMN: default mode network; fMRI: functional magnetic resonance imaging; HC: healthy controls; ICA: independent component analysis; MFG: middle frontal gyrus; MwA: migraine with aura; MwoA: migraine without aura; NCF: nucleus cuneiforms; PAG: periaqueductal gyrus; ReHo: regional homogeneity; RS-FC: resting state functional connectivity; SMN: sensory-motor network; SN: salience network; SSC: somatosensory cortex; SFG: spatial frequency of grating; VN: visual network.

**Table 2 brainsci-11-00234-t002:** FMRI case control connectivity studies employing pain-related stimulation.

Year	Authors	Subjects	Condition (Ictal-Intercritical)	Methods	Main Results	Significance
2017	Mehnert et al. [29]	MWoA = 8 CM = 46		fMRI to gaseous ammonia stimuli	↑ activation in PAG, left crus I, the latter was less connected with left thalamus, bilateral occipital areas, and the right fusiformis gyrus.	This study emphasizes the important role of the cerebellum in nociception.
2017	Hebestreit & May [71]	Episodic = 13 Chronic = 6	Interiritical	Trigeminal fMRI recording before and after administration of metoprolol 75mg	No central effects. In an exploratory analysis, metoprolol slightly enhanced hypothalamic activity	Metoprolol seems to act peripherally, with negligible central effects
2018	Schulte et al. [63]	MWoA = 18 CM = 17 HC = 19	Ictal/Intercrticial	fMRI to visual stimuli	In chronic patients ↑ activation in the spinal trigeminal nuclei (↑during the headache) and right superior colliculus.	Abnormal visual–nociceptive integration at the brainstem level during the headaches
2020	Ziegeler et al. [72]	Episodic = 12 Chronic = 15	Interictal	Trigeminal fMRI recording before and after administration of erenumab 70mg	Decreased activation in several brain areas after erenumab. Only responders show reduced hypothalamic activation.	The efficacy of Erenumab may not be confined exclusively to the periphery

Legend: MWoA: migraine without aura: CM: Chronic Migraine; fMRI: functional magnetic resonance imaging: PAG: Periacqueductal grey gyrus.

**Table 3 brainsci-11-00234-t003:** Resting state and Pain stimulation-related EEG/MEG connectivity changes.

Year	Authors	Subjects	Condition (ictal-intercritical)	Methods	Main results	Significance
2016	Cao et al. [79]	50 MWA 20 HC	Ictal-pre-post ictal	EEG power and coherence analyses Resting state	↓EEG power and coherence in inter and ictal phases	RS EEG power and connectivity fluctuate across migraine phases
2016	Wu et al. [38]	13 MWA 10 MWoA 23 HC	Intercritical	MEG Graph analysis Resting state	↑functional connectivity and connections stenght and path lenght in slow wave and fast bands	Functional connectivity impaired in low- and high-frequency ranges, possible sign of brain reorganization.
2017	de Tommaso et al. [12]	19 MWA 19 MWoA 11 HC	Intercritical	EEG 65 channels FMRI Granger causality Graph analysis Resting state Visual stimulation	Different information flow among MWoA, MWA and controls in resting state. Different brain networking in MWA in occipital regions, according to FMRI results	Phenotypical differences in neuronal networks organization at occipital level, related to aura symptoms perception
2020	Frid et al. [78]	22 MWoA 30 MWA	Intercritical	EEG 26 channels Machine Learning on Functional Connectivity Resting State	↓ connectivity in theta band in MWA, in selected brain areas in Default Mode and Resting State Networks	↓ functioning of the DMN in migraine with aura.
2020	Nieboer et al. [81]	24 MWA 24 HC	Intercritical	MEG Functional connectivity between 78 cortical brain regions phase lag index Resting State	↑betweeness centrality in higher frequency bands in patients with longer disease duration	Specific brain areas have altered topological roles in a frequency-specific manner
2020	Hsiao et al. [82]	30 MWoA 27 HC	Intercritical	MEG functional connectivity at 2 to 59 Hz in pain-related cortical region Correlation with pain threshold Resting State	Pain threshold inversely correlated with gamma oscillation in C, no correlation in MWoA	Lack of correlation between functional connectivity in pain related structures and pain sensitivity in migraine
2015	de Tommaso et al. [13]	31 MWoA 19 HC	Intercritical	EEG 65 channels Synchronization entropy Granger causality Habituation Laser stimulation	↑ information flow between the bilateral temporal-parietal and the frontal regions in MWoA Correlation with laser evoked potentials dis-habituation	Change in brain function within pain related cortical areas in migraine
2019	Ren et al. [53]	22 MWoA 22 HC	Intercritical	MEG Graph theory Electrical stimulation	↑ functional connectivity in the high-frequency band between the sensory cortex and the frontal lobe in migraine	Aberrant connections from the somatosensory cortex to the frontal lobe.
2020	Bassez et al. [83]	23 MWoA 20 HC	Intercritical	EEG 65 channels Dynamical Causal Modelling Laser stimulation	↑ relationships between lS1 and the thalamus, in both directions, with a lack of progressive habituation of connection strengths from the thalamus to all somatosensory areas	Disrupted thalamus-cortical networks dynamic, coherent with reduced habituation to painful stimuli

Legend: MWoA Migraine Without Aura: MWA: Migraine With Aura; HC: Healthy Controls; RS: Resting State: EEG: ElectroEncephalogram; MEG: Magnetoencephalogram.

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
