# Peer review of "Pain-Related Brain Connectivity Changes in Migraine: A Narrative Review and Proof of Concept about Possible Novel Treatments Interference"

_brainsci, 2021, doi:10.3390/brainsci11020234_

Round 1

Reviewer 1 Report

This is a narrative review of recent (past 5 years) case-control functional imaging studies (fMRI and EEG/MEG) in persons with migraine. The intended aim is “to provide for a mechanistic hypothesis about the possible interference on the migraine dynamic model from the novel CGRP-related peripheral approach. “ This objective is also inferred by the title, “Pain-related brain connectivity changes in migraine: a narrative review and proof of concept about possible novel treatments interference”.

The review, written by authors who are well recognized authorities in this field, is thorough and well-referenced. The accompanying tables are a particularly valuable resource for persons interested in this topic. My criticisms, as listed below, are meant to be constructive.

---The Abstract mention migraine triggers, a phenomenon that may also connect the peripheral environment to the CNS. The Abstract states, “Studies on FC have shown that cortical processes, in  the neurolimbic pain network, are likely to be prevalent for triggering attacks, in response  to predisposing factors, and that these lead to a demodulation of the subcortical areas, at  the basis of migraine maintenance”.  The meaning of this long sentence is not clear. I take it that cortical processes is the subject, in which case these cortical processes are likely to be prevalent for triggering attacks and to demodulate subcortical areas, a key event in continuing the migraine headache. Is this correct?

--The Introduction briefly summarizes current understanding of the pathophysiology of migraine. It would be helpful to expound on evidence-based concepts and make clear what are gaps in knowledge.

--The authors set up their Aim by mentioning the role of CGRP in migraine-related inflammation and that peripheral acting CGRP receptor antagonists interrupt headache generation. They go onto to state, “The great challenge for the migraine approach is the understanding of the theoretic basis of the long-term effect of peripheral interaction via CGRP inhibition on the complex central dysfunction which could account for migraine inclusion within the “oscillopathies”. This sentence typifies many of those within the paper, where the meaning in obfuscated by syntax and verbosity. The concepts addressed in this are complicated enough; plain language will definitely improve readability.

--The closing sentence of the Introduction “The complexity of brain function is based on dynamic relationships among cortical and subcortical areas, which enable the brain to adapt itself to different physiological and pathological conditions, a feature possibly affected in the migraine brain” does not clearly relate as I read it to the following aim of connecting the central mechanisms of migraine gleaned from functional imaging to peripheral mechanisms, which are susceptible to treatment with CGRP antagonists.

-The Introduction contains the sentence, “It is unclear how a typical migraine attack is triggered - this is one of the most important unanswered questions in migraine neuroscience,” but yet triggers are not clearly addressed in the text or General Remarks at the end, which I expected. Some of the studies in the text discuss aura, which usually precedes headache, but if the studies shed light on the mechanisms of factors triggering migraine, this should be stated explicitly.

-Methods. How many case control studies were evaluated? How many excluded and for what reasons?

-FMRI (lines 125-127): Only ictal CA was evaluated in ref 24, are the authors inferring that atypical rs-fc of brainstem descending modulatory pain regions with other brainstem and higher order pain-modulating regions is also associated with interictal allodynia?

-FMRI (lines 159-161): What are the reference for this and is chronicization a word?: “In the context of subcortical structures, recent observations have supported the different role of the left or right amygdala in migraine pathophysiological mechanisms, suggesting the involvement of the left amygdala in migraine genesis as well as of the right amygdala in migraine chronicization.”

-FMRI (line 322): mentions Table 3, but referring I believe to Table 1

-FMRI (lines 373-377): sentence is unclear. Please rewrite.

-General Remarks. Lines 499-500 echo lines 23-25 in the Abstract, but I don’t know where this was explicitly discussed in the text.

-Perspectives. It is not until the last 4 paragraphs of the paper that the main aim is of the review is brought to the fore. It would be helpful to refer to the aim when appropriate throughout the narrative review.

-The very important last sentence definitely need to be reworked for clarity

Major points on language

-Edit for readability. Multiple long sentences where meaning is obscured by syntax.

Minor points on language

-Substitute ‘interictal’ for ‘intercritical’

-Find a different word for ‘subtend(ing)’, used 6 times, which is obscure except in the field of mathematics, and doesn’t seem correct in this context.

-Find a different term for ‘anti-correlated’

-Avoid overuse of phrases like ‘it is noteworthy’ and ‘interestingly’ (used 5 times).

-Spell out all abbreviations when first used, even though they are in the legend of the tables

-Edit for grammatical errors of which there are several

Author Response

Response to reviewer
Reviewer #1

This is a narrative review of recent (past 5 years) case-control functional imaging studies (fMRI and EEG/MEG) in persons with migraine. The intended aim is “to provide for a mechanistic hypothesis about the possible interference on the migraine dynamic model from the novel CGRP-related peripheral approach. “ This objective is also inferred by the title, “Pain-related brain connectivity changes in migraine: a narrative review and proof of concept about possible novel treatments interference”. The review, written by authors who are well recognized authorities in this field, is thorough and well-referenced. The accompanying tables are a particularly valuable resource for persons interested in this topic. My criticisms, as listed below, are meant to be constructive.

Question #1: The Abstract mention migraine triggers, a phenomenon that may also connect the peripheral environment to the CNS. The Abstract states, “Studies on FC have shown that cortical processes, in  the neurolimbic pain network, are likely to be prevalent for triggering attacks, in response  to predisposing factors, and that these lead to a demodulation of the subcortical areas, at  the basis of migraine maintenance”. The meaning of this long sentence is not clear. I take it that cortical processes is the subject, in which case these cortical processes are likely to be prevalent for triggering attacks and to demodulate subcortical areas, a key event in continuing the migraine headache. Is this correct?”
Response #1: Thank you for the constructive and useful criticism. The requested changes are outlined in yellow color. The abstract has been modified and the phrase reconstructed as A disrupted FC in the neurolimbic pain network could favor the triggering of the attacks through the activation of those subcortical areas which are the basis of migraine maintenance.”

Question #2: The Introduction briefly summarizes current understanding of the pathophysiology of migraine. It would be helpful to expound on evidence-based concepts and make clear what are gaps in knowledge.
Response #2: We added a short comment on main migraine pathophysiology gaps: However, there are still several gaps in migraine pathophysiology understanding. The attack generation modality in migraine without aura (e.g., the role of the CSD) and – consequently - the mechanism of antidromic trigeminal fibers activation, the real mechanism of the most of preventative drugs on such a complex causative mechanism and the long-term changes within migraine brain are still unclear points.”

Question #3: The authors set up their Aim by mentioning the role of CGRP in migraine-related inflammation and that peripheral acting CGRP receptor antagonists interrupt headache generation. They go onto to state, “The great challenge for the migraine approach is the understanding of the theoretic basis of the long-term effect of peripheral interaction via CGRP inhibition on the complex central dysfunction which could account for migraine inclusion within the “oscillopathies”. This sentence typifies many of those within the paper, where the meaning in obfuscated by syntax and verbosity. The concepts addressed in this are complicated enough; plain language will definitely improve readability.
Response #3: We thank the reviewer for the suggestion. We have conducted an in depth revision of English language. The sentence has been reconstructed as follow: “Furthermore, the great challenge for the migraine approach is the longstanding effect of peripheral interaction via CGRP inhibition on the complex dysfunction in brain dynamic. In fact, relationships among cortical and subcortical areas are possibly affected in the migraine brain [20] and still unclearly influenced by peripheral CGRP inhibition.”

Question #4: The closing sentence of the Introduction “The complexity of brain function is based on dynamic relationships among cortical and subcortical areas, which enable the brain to adapt itself to different physiological and pathological conditions, a feature possibly affected in the migraine brain” does not clearly relate as I read it to the following aim of connecting the central mechanisms of migraine gleaned from functional imaging to peripheral mechanisms, which are susceptible to treatment with CGRP antagonists.
Response #4: We have rephrased and simplified in “the great challenge for the migraine approach is the longstanding effect of peripheral interaction via CGRP inhibition on the complex dysfunction in brain dynamic. In fact, relationships among cortical and subcortical areas are possibly affected in the migraine brain [20] and still unclearly influenced by peripheral CGRP inhibition.”

Question #5: The Introduction contains the sentence: “It is unclear how a typical migraine attack is triggered - this is one of the most important unanswered questions in migraine neuroscience”, but yet triggers are not clearly addressed in the text or General Remarks at the end, which I expected. Some of the studies in the text discuss aura, which usually precedes headache, but if the studies shed light on the mechanisms of factors triggering migraine, this should be stated explicitly.
Response #5: Considering that migraine triggers is not the primary point of the present review, we preferred to remove the sentence.

Question #6: Methods. How many case control studies were evaluated? How many excluded and for what reasons?
Response #6: We added these information: “By using “migraine, fMRI, connectivity, pain” as key words we obtained 96 references, including 11 reviews and 7 animal studies. Thus, we selected 23 studies among them because 55 studies included works with neuroimaging methods different from FMRI and not case control studies. Moreover, searching for “migraine, EEG, MEG, connectivity, pain” led us to 10 results, which were here included with the exclusion of 1 review.

Question #7: fMRI (lines 125-127): Only ictal CA was evaluated in ref 24, are the authors inferring that atypical RS-FC of brainstem descending modulatory pain regions with other brainstem and higher order pain-modulating regions is also associated with interictal allodynia?
Response #7: We thank the reviewer for the opportunity of clarifying this issue. No, authors found a correlation with ictal allodynia, though they evaluated it also in the interictal period. However, we have clarified the meaning of the pinpointed sentence as follows: “Altogether, the findings support dysfunctional top-down endogenous pain-inhibitory mechanisms, detectable also during intercritical period, particularly relevant in migraine patients experiencing CA.”

Question #8: fMRI (lines 159-161): What are the reference for this and is chronicization a word?: “In the context of subcortical structures, recent observations have supported the different role of the left or right amygdala in migraine pathophysiological mechanisms, suggesting the involvement of the left amygdala in migraine genesis as well as of the right amygdala in migraine chronicization.”
Response #8: Thank you, we omitted to report “Chen Z, Chen X, Liu M, Dong Z, Ma L, Yu S. Altered functional connectivity of amygdala underlying the neuromechanism of migraine pathogenesis. J Headache Pain. 2017 Dec;18(1):7. doi: 10.1186/s10194-017-0722-5. Epub 2017 Jan 23. PMID: 28116559; PMCID: PMC5256627.” that it has been added to the reference list.
We replaced the term with “chronic evolution”.

Question #9: fMRI (line 322): mentions Table 3, but referring I believe to Table 1
Response#9: We corrected, thank you.

Question #10: fMRI (lines 373-377): sentence is unclear. Please rewrite.
Response #10: We have rewritten as follows : “The images accompanying these articles showed that, apart from the hypothalamus, the primary visual areas are active during both the pre-ictal and ictal phases [8,64], as already demonstrated by other studies using PET scanning [67].”

Question #11: General Remarks. Lines 499-500 echo lines 23-25 in the Abstract, but I don’t know where this was explicitly discussed in the text.
Response #11: The Reviewer is right. Now, to improve clarity, we have revised the sentence as follows: “Moreover, migraine appears a brain pervasive condition, with a huge involvement of aberrant cortical processes, which may lead to migraine attacks susceptibility.”

Question #12: Perspectives. It is not until the last 4 paragraphs of the paper that the main aim is of the review is brought to the fore. It would be helpful to refer to the aim when appropriate throughout the narrative review.
Response #12: The reviewer is completely right. The real aim of the study was better clarified in the introduction: “This review intends to provide a general view about pain-related connectivity fea-tures in migraine, in order to propose a preliminary mechanistic hypothesis about the possible interference of the novel CGRP-related peripheral approach.”

Question #13: The very important last sentence definitely need to be reworked for clarity
Response #13: We have rephrased as follows: “The complex neuronal dynamic architecture at the base of migraine disease is worthy of an integrated approach, especially combining neurophysiological and neuroimaging methods. However, future long term efficacy studies on CGRP-mAbs, coupled with neurophysiological evidence of possible interference on brain connectivity features, could possibly confirm the hypothesis that a specific and prolonged action on the trigemino-vascular system, though limited to the periphery, has a neuromodulatory effect on the central, still elusive, mechanisms of migraine.”

Question #14: Major points on language. Edit for readability. Multiple long sentences where meaning is obscured by syntax.
Response #14: We thank the reviewer for the suggestion. We have conducted a further in deep revision of English language.

Question #15:  Minor points on language. Substitute ‘interictal’ for ‘intercritical’. Find a different word for ‘subtend(ing)’, used 6 times, which is obscure except in the field of mathematics, and doesn’t seem correct in this context. Find a different term for ‘anti-correlated’. Avoid overuse of phrases like ‘it is noteworthy’ and ‘interestingly’ (used 5 times). Spell out all abbreviations when first used, even though they are in the legend of the tables.
Response #15: We thank the reviewer for the observations. We have reviewed the manuscript following the suggestions. Specifically, we have substituted ‘interictal’ for ‘intercritical’, replaced the word “subtend(ing)” with others with more appropriate meaning through the entire manuscript, and the word “anti-correlated” with “inverse correlation”. We spelled out all abbreviations. Finally, we conducted a further in deep revision of English language.

Reviewer 2 Report

The article summarizes and discusses the results of functional imaging studies on migraine, and clarifies the alteration of functional connectivities among cerebral structures in different migraine states.

1. Major:

1.1The structure of this article is unclear. The part on the imaging analysis is not structured in the review. Please restructure it, especially for the “Pain stimulation-related FMRI connectivity changes” part. For example, it could divide the different analysis methods of fMRI data into different sections.

1.2 In the introduction part, it describes the molecular biological findings, such as CGRP, which is later summarized in the part6. It seems to have no relation to part5. The major part of the article focused on the imaging findings at different stages of the migraine and is not related to the introduction part, either.

1.3 The heading of the article is about “Pain-related brain connectivity changes in migraine: a narrative review and proof of concept about novel treatments interference.”. The article combines two topics (imaging findings and treatment interference). However, the article is not clarified in focus. The main part of the article mainly states the changes in the results from the imaging analysis on migraine patients, if the changes in imaging related to novel treatment were elaborated in the article, it would make the topic clearer. 

2 Minor:

2.1.The functional connectivity based on the fMRI is from the BOLD signal and functional relationship among different cerebral regions, which can be used to generate the functional matrix. Then, the graphic analysis applies to the analysis on it. It should be noticed that these connections are functional. In Line 305-307, “According to the graph theory analysis, the brain network is investigated as a set of grey matter regions (nodes) structurally connected by white matter paths (edges).”, which is the definition of the structural connectome mostly based on the DTI but not for the fMRI.

2.2. Some English mistakes in the article, please go through and check them again. For instance, Line 297-298 “whether a “specific FC 297 brain map” is able to identify a migraine patient or a HC.”, it should be “an HC”. Line 433 “One of the first MEG connectivity study in migraine described”, it should be “ studies”.

2.3 Some abbreviations have no corresponding full names in the article, such as PO and VPL

Author Response

Reviewer #2

The article summarizes and discusses the results of functional imaging studies on migraine, and clarifies the alteration of functional connectivities among cerebral structures in different migraine states.

Major:
Question #1: The structure of this article is unclear. The part on the imaging analysis is not structured in the review. Please restructure it, especially for the “Pain stimulation-related FMRI connectivity changes” part. For example, it could divide the different analysis methods of fMRI data into different sections.
Response #1: Thank you for your useful comments. The requested changes are outlined in green. According to the Reviewer suggestion, now we have subdivided the pinpointed pain in two sections: “Intercritical investigation” and “Prodromal and headache phase investigation”. The FMRI methods are divided in sub-headings

Question #2: In the introduction part, it describes the molecular biological findings, such as CGRP, which is later summarized in the part 6. It seems to have no relation to part 5. The major part of the article focused on the imaging findings at different stages of the migraine and is not related to the introduction part, either. Question #3: The heading of the article is about “Pain-related brain connectivity changes in migraine: a narrative review and proof of concept about novel treatments interference”. The article combines  two topics (imaging findings and treatment interference). However, the article is not clarified in focus. The main part of the article mainly states the changes in the results from the imaging analysis on migraine patients, if the changes in imaging related to novel treatment were elaborated in the article, it would make the topic clearer.
Response #2 and #3: We have just modified the main aim of the study , in accord to reviewer 1: Now, following the Reviewer’s requirement, we have added a subsection in the “Pain stimulation-related FMRI connectivity changes” section about effects of migraine preventives, including novel monoclonal antibodies, on trigeminal fMRI. This section was titled “Effects of preventive drugs”. We have updated Table 2 too.

Minor:
Question #4: The functional connectivity based on the fMRI is from the BOLD signal and functional relationship among different cerebral regions, which can be used to generate the functional matrix. Then, the graphic analysis applies to the analysis on it. It should be noticed that these connections are functional. In Line 305-307, “According to the graph theory analysis, the brain network is investigated as a set of grey matter regions (nodes) structurally connected by white matter paths (edges).” which is the definition of the structural connectome mostly based on the DTI but not for the fMRI.
Response  #4: We thank the reviewer for the opportunity of clarifying this issue.  We have better clarified the different methods that can be applied to study the brain connectome and how studies using a DTI approach largely prevail in migraine literature. We have added the following sentence:
“The brain connectome could be investigated both using RS-fMRI and diffusion tensor imaging approaches. However, at the moment, in the migraine literature, we find only data related to the second methodological approach.”

Question #4: Some English mistakes in the article, please go through and check them again. For instance, Line 297-298 “whether a “specific FC 297 brain map” is able to identify a migraine patient or a HC.”, it should be “an HC”. Line 433 “One of the first MEG connectivity study in migraine described”, it should be “ studies”.
Response #4: we thank the reviewer for the suggestion. We conducted an in depth revision of English language.

Question #5: Some abbreviations have no corresponding full names in the article, such as PO and VPL
Response #5: added in full.

Date of this review: 22 Jan 2021

Round 2

Reviewer 2 Report

The readability of this article after revision is improved and the structure is more reasonable compared to the previous version.
Minor questions:

1. “A novel advanced MRI methodological approach emerged to explore the structural neural substrates enabling functional communication within the brain “connectome”.” The fMRI connectome is not structurally but functionally based. It is the matrices based on the correlation of the BOLD signal changes between every two regions in a period of time, which are not anatomically or structurally connections based.

2. In lines 214 to 258, a large introduction to the RS-FC and network in fMRI, which is difficult to connect with the content after line 259 "altogether,...". I suggest to briefly describe the RS-FC and network in fMRI. Regarding "stressors trigger migraine attacks", it can be taken as a perspective and summarized in this paragraph。

Author Response

The readability of this article after revision is improved and the structure is more reasonable compared to the previous version.
Minor questions:

Thank you for the good opinion about our revision, and your request for minor amendment.  

           Question 1

  1. “A novel advanced MRI methodological approach emerged to explore the structural neural substrates enabling functional communication within the brain “connectome”.” The fMRI connectome is not structurally but functionally  It is the matrices based on the correlation of the BOLD signal changes between every two regions in a period of time, which are not anatomically or structurally connections based.

Replay to question 2: We deleted “structural”

Question 2

  1. In lines 214 to 258, a large introduction to the RS-FC and network in fMRI, which is difficult to connect with the content after line 259 "altogether,...". I suggest to briefly describe the RS-FC and network in fMRI. Regarding "stressors trigger migraine attacks", it can be taken as a perspective and summarized in this paragraph。

Replay to question 2

We have consistently simplified the paragraph. The requested changes are outlined in green
